# Fabrication of PCL Scaffolds by Supercritical CO_2_ Foaming Based on the Combined Effects of Rheological and Crystallization Properties

**DOI:** 10.3390/polym12040780

**Published:** 2020-04-02

**Authors:** Chaobo Song, Yunhan Luo, Yankai Liu, Shuang Li, Zhenhao Xi, Ling Zhao, Lian Cen, Eryi Lu

**Affiliations:** 1Shanghai Key Laboratory of Multiphase Materials Chemical Engineering, School of Chemical engineering, East China University of Science and Technology, Shanghai 200237, China; scbecust@126.com (C.S.); yunhanluo_ecust@163.com (Y.L.); liuyankai.ecust@gmail.com (Y.L.); zhaoling@ecust.edu.cn (L.Z.); cenlian@hotmail.com (L.C.); 2School of Medicine, Shanghai Jiaotong University, Shanghai 200127, China; lishuangefg@126.com; 3College of Chemistry and Chemical Engineering, Xinjiang University, Urumqi 830046, China

**Keywords:** polycaprolactone (PCL), supercritical carbon dioxide (scCO_2_), tissue engineering, rheology, crystallization

## Abstract

Polycaprolactone (PCL) scaffolds have recently been developed via efficient and green supercritical carbon dioxide (scCO_2_) melt-state foaming. However, previously reported gas-foamed scaffolds sometimes showed insufficient interconnectivity or pore size for tissue engineering. In this study, we have correlated the thermal and rheological properties of PCL scaffolds with their porous morphology by studying four foamed samples with varied molecular weight (MW), and particularly aimed to clarify the required properties for the fabrication of scaffolds with favorable interconnected macropores. DSC and rheological tests indicate that samples show a delayed crystallization and enhanced complex viscosity with the increasing of MW. After foaming, scaffolds (27 kDa in weight-average molecular weight) show a favorable morphology (pore size = 70–180 μm, porosity = 90% and interconnectivity = 96%), where the lowest melt strength favors the generation of interconnected macropore, and the most rapid crystallization provides proper foamability. The scaffolds (27 kDa) also possess the highest Young’s modulus. More importantly, owing to the sufficient room and favorable material transportation provided by highly interconnected macropores, cells onto the optimized scaffolds (27 kDa) perform vigorous proliferation and superior adhesion and ingrowth, indicating its potential for regeneration applications. Furthermore, our findings provide new insights into the morphological control of porous scaffolds fabricated by scCO_2_ foaming, and are highly relevant to a broader community that is focusing on polymer foaming.

## 1. Introduction

Polycaprolactone (PCL), a semi-crystalline polyester with outstanding mechanical properties, favorable biocompatibility and proper degradability, is widely applied as a promising biomedical material, especially for tissue engineering [1,2,3,4]. In tissue engineering, the scaffold is considered as a structural template that provides an ideal environment for the adhesion, proliferation, differentiation and migration of cells. To achieve these biological functions, previous works have generally recommended that ideal scaffolds should have macropores (100–1000 µm), as well as high interconnectivity as possible [5,6]. Numerous methods have been developed to fabricate porous structures, including solution casting/salt leaching, electrospinning [1,2], freeze drying [3], phase separation [4,7] and 3D printing [8]. However, these methods possess some limitations, such as the use of organic solvents, which are nearly impossible to remove completely after scaffolds being fabricated, and/or elevated temperature, which always results in accelerating polymer degradation [9]. Recently, Ismail et al. applied a relatively safe in situ photocrosslinked electrospinning method to prepare fibrous scaffolds under mild processing conditions [10]. However, the prepared scaffolds in their study showed a limited pore size of about ten microns, which may have negative effects on the ingrowth of cells. As a pore-forming method, supercritical carbon dioxide (scCO_2_) foaming has unique advantages, such as environmental friendliness, non-organic solvents, non-flammability and a mild supercritical state (critical temperature = 31.1 °C, critical pressure = 7.38 MPa) [11,12]. More importantly, the plasticization effect of scCO_2_ can lower the melting point of crystalline polymers, as well as the processing temperature [13]. Additionally, simple and efficient depressurization at a relatively low temperature can lead to the formation of controllable porous morphology, including pore size, porosity and interconnectivity [14,15,16,17]. Considering the above-mentioned benefits, the scCO_2_ foaming strategy is a promising alternative for the preparation of PCL scaffolds. However, it is still a big challenge to fabricate stable PCL scaffolds with an ideal porous morphology by pure scCO_2_ foaming [18,19,20,21].

In recent years, efforts have been made to investigate the influence of processing conditions (e.g., temperature, pressure and depressurization rate) on the porous morphology of PCL scaffolds. With the increase of foaming temperature, the foaming process shifts from a solid-state to a melt-state, which generally results in a relatively larger pore size and higher porosity of PCL foams [22]. However, excessive high foaming temperature can also lead to a visible collapse and dense phase of scaffolds [22,23,24,25]. Meanwhile, it has been reported that PCL foams tend to generate favorable macropores when processed under low foaming pressure [23,24,26]. In addition, our previous study demonstrated that a high depressurization rate is beneficial to achieve high interconnectivity [27]. However, other reports have indicated that PCL samples with different intrinsic properties always possess varied foaming results, and some scaffolds showed insufficient pore size and interconnectivity for the usage of tissue engineering due to the improper selection of PCL samples. For example, Karimi, M. et al. reported that PCL scaffolds (120 kDa) prepared at the foaming conditions (T = 40 °C, P = 100 bar and venting time = 10 bar/s) showed a favorable pore size around 260 μm [21], while Chen, C.X. et al. reported that PCL scaffolds (80 kDa) gained an insufficient average pore size around 60 μm and interconnectivity of below 75% under similar conditions [22]. The insufficient pore size and interconnectivity may hinder the ingrowth of cells and the transition of nutrition and waste. In other studies, an additional particle-leaching was carried out to improve the pore size and interconnectivity [18,19]. However, this two-step method extends the processing period, resulting in low efficiency, and the possible residual chemicals might cause a negative biological response, such as inflammation. Hence, to fabricate PCL scaffolds with a desired highly interconnected and macroporous structure by pure foaming strategy, the intrinsic properties of PCL should be carefully considered.

Generally, rheological and crystallization characteristics are two main intrinsic properties of polymers, which play a key role in the foaming process. First, rheological properties can modulate foaming results by greatly influencing the growth and fracture of pores, and samples with a high melt strength always hinder foam expansion and the fracture of pores [20,28]. For example, in the studies of poly(lactic-co-glycolic acid) (PLGA) foaming, samples with lower melt strength possess an increased pore size but decreased porosity, owing to the improved growth of pores and the fracture of pore walls [15,29]. Second, for semi-crystalline polymers, crystallization behaviors also significantly influence the foaming results [30,31,32]. In a melt-state foaming, crystals are generated by the temperature decreasing after depressurization. The stiffness of polymers can be improved significantly to promote the solidification of pores due to the entangled polymer molecules through crystals [33]. The resulting promoted solidification always constrains the deformation of porous structures to form small and closed pores. Furthermore, the promoted solidification can also suppress the collapse of the pore to improve the foamability, which is essential for a matrix with a low melt strength to maintain its relative integral porous structure after foaming. Thus, both crystallization and rheological properties need to be considered for thoroughly understanding the final porous structure of PCL scaffolds. 

In this study, we tried to correlate the final morphology of PCL scaffolds with their rheological and crystallization behaviors, and specifically aimed to clarify the proper intrinsic properties for developing scaffolds with favorable highly interconnected and macroporous morphology. To achieve this, PCL samples with a different MW (27, 54, 100 and 219 kDa in weight-average molecular weight) were used with the expectation to obtain varied crystallization and rheological properties, and a melt-state foaming with mild temperature, low pressure and rapid depressurization was also carried out, as these foaming conditions were reported to be a beneficial aspect for the improvement of pore size and interconnectivity. 

## 2. Materials and Methods

### 2.1. GPC Measurements

Polycaprolactone (PCL) samples with different molecular weights were purchased from Jinan Daigang Biomaterial Company (Shandong, China). A gel permeation chromatography (GPC50, PL, UK) was applied to determine the molecular weight (MW) of these PCL samples. The GPC is equipped with a refractive index (RI) detector and two non-polar organic columns (PLgel 5 μm MIXED-C, 300 × 7.5 mm). The molecular weights were determined by using a dn/dc of 0.113 mL/g, which was calculated from the RI detector response with the assumption of 100% mass recovery through the column. Prior to testing, PCL samples were dissolved into tetrahydrofuran (THF, analytical grade, Honeywell) at a concentration of 20 mg/mL. The flow rate of the mobile phase (THF) was 1 mL/min, and the injection volume was 100 mL. GPC results were processed with software (Cirrus GPC Version 3.4) by using narrow standards. The resulting weight-average molecular weight (*M_w_*), number-average molecular weight (*M_n_*) and polydispersity index (PDI) are shown in Table 1, and the MW distribution is shown in Figure 1. Results indicated that the PDI of these samples were around 1.7, and the *M_w_* and *M_n_* of adjacent samples increased to approximately two folds. PCL samples were consecutively labeled as PCL-1, PCL-2, PCL-3 and PCL-4 with the increase of MW respectively.

### 2.2. Foaming Process

The PCL scaffolds were fabricated by a rapid depressurization batch foaming process [27]. A custom-made, lidless PTFE mold (four wells; diameter: 15 mm; height: 20 mm) was used to hold PCL samples, and the construction of mold-samples was then placed in a high-pressure vessel (height: 80 mm; diameter: 50 mm) which was sealed afterwards. The vessel temperature was then monitored and controlled to 50 °C by a super thermostat (CH-1015, Shanghai Hengping Instrument and Meter factory, Shanghai, China) with an accuracy of 0.1 °C. High pressure pipes and valves (SS-4SKPS8MM, Swagelok, Ohio, USA) were applied to connect the vessel and gas supply system. A syringe pump (DZB-1A, Beijing Satellite Instrument Co., Beijing, China) was applied to pump the CO_2_ into the high-pressure vessel. The vessel was first swept with CO_2_ three times at approximately 1 bar and 30 seconds each time to squeeze the air within the vessel. Then, CO_2_ was pumped in to reach the set pressure of 10 MPa with an accuracy of 0.1 MPa. The set temperature and pressure were maintained for 120 min to achieve the equilibrium of CO_2_ absorption. Finally, the venting valve was rapidly opened to depressurize the vessel to atmosphere and trigger the foaming of PCL. Finally, the vessel was open, and all samples were removed for further characterization.

### 2.3. Thermal Analyses

The thermal analyses of PCL samples with a different MW (shown in Table 1) were carried out by a differential scanning calorimetry (DSC, NETZSCH DSC204 HP, Bavaria, Germany) under an N_2_ atmosphere at 1 bar. The samples were first heated to 100 °C with a heating rate of 20 °C/min, and maintained at the terminal temperature for 5 min to completely remove the thermal history of all samples. Then, to analyze the crystallization properties, the samples were cooled down to 20 °C with a cooling rate of 5 °C/min. After maintaining for 5 min to determine the melting properties, samples were again heated to 100 °C with a heating rate of 10 °C/min. 

### 2.4. Rheological Measurements

The shear dynamic rheology behaviors of PCL samples were measured by a rheometer (TA-ARES, TA Instruments, New Castle, DE, USA) with a 25 mm parallel plate for frequency sweep. Prior to testing, PCL samples were first hot pressed into slices with a diameter of 25 mm and a height of 1 mm at 100 °C, 10 MPa for 2 min. Dynamic frequency sweep tests were carried out at 80 °C from 0.1 to 1000 rad/s. The oscillation strain was fixed at 5% to ensure all PCL samples in the linear viscoelastic region. The complex viscosity (*η**), storage modulus (*G*′) and loss modulus (*G*″) were measured as a function of frequency with the range of 0.1~1000 rad/s.

### 2.5. Scanning Electron Microscopy (SEM)

A scan electron microscopy (SEM, JSM-6360LV, JEOL Ltd, Tokyo, Japan) was applied to investigate the porous morphology of prepared scaffolds. Samples were first freeze-fractured in liquid nitrogen and then fixed to SEM stubs using carbon tape. The samples were also sputter-coated with Pd (Palladium) before SEM observation. The SEM images were then analyzed by Image Pro-plus software (Media Cybernetics, Silver Spring, MD, USA) to calculate the average pore size by counting the pores, with more than 200 pores counted for each sample. 

### 2.6. Measurement of Porosity

According to ASTM D792-0046, the porosity of PCL samples was measured and calculated by the following equations
(1)$$\rho_{f}=\frac{a}{a+w-b}\rho_{water}$$
(2)$$Porsity=\frac{\frac{\rho_{0}}{\rho_{f}}-1}{\frac{\rho_{0}}{\rho_{f}}}\times 100$$
where *a*, *b* and *w* represent the apparent mass of the sample in air, the apparent mass of the sample and sinker in water and the apparent mass of the sinker in water, respectively.

### 2.7. Measurement of Interconnectivity

First, the PCL samples were fractured in liquid nitrogen to avoid any possible damage. An Ultrapycnometer 1000 (Quantachrome Instruments, Boynton Beach, FL, USA) single station automatic gas pycnometer was applied to measure the closed volume (*V_c_*) of samples. The interconnectivity of PCL scaffolds was calculated by the following equations
(3)$$V_{p}=\frac{a}{\rho_{f}}$$
(4)$$Interconnectivity=\frac{V_{p}-V_{c}}{V_{p}\times Porosity}\times 100\%$$
where *V_p_* represents the total volume of the foamed sample. Interconnectivity was calculated as its definition of the volume fraction of interconnected pores in the entire sample.

### 2.8. Measurements of Compressive Properties

First, samples were cut into slices (diameter: 15 mm, height: 8 mm). A universal testing machine (AG-2000A, Shimadzu, Kyoto, Japan) was applied to perform the compressive tests at a cross-head speed of 1 mm/min in the air under ambient conditions. The stress was calculated according to the original cross-sectional area of the samples, and determined as a function of the strain. The yield compressive strength was recorded based on the stress-strain curves, and Young’s modulus was calculated at 10% strains. Five samples were tested for each group.

### 2.9. In Vitro Cell Tests

(1) Seeding cells: Prior to cell seeding, PCL scaffolds were first cut into 2 mm × 12 mm × 12 mm and then *γ*-sterilized at 25K rad for 8 min. After that, the sterilized scaffolds were placed in 24-well plates. To avoid the leakage of the cells after seeding, custom-made stainless steel rings (height: 2 mm; outer diameter: 11 mm; thickness: 1 mm) were then carefully fixed onto the samples. L929 cells (purchased from Cell Resource Center, IBMS, CAMS/PUMC, Shanghai, China) were carefully seeded onto the samples, with a density of 2 × 10^4^ cells per sample. After seeding, Dulbecco’s Modified Eagle Medium (DMEM, HyClone, Logan, UT, USA) containing 10 % Fetal Bovine Serum (FBS, HyClone, Utah, USA) was applied to nourish the seeded L929 cells. The cell-scaffold constructions were then cultured in a human-made atmosphere of 95%air/5%CO_2_ (*v*/*v*) at 37 °C, and the culture mediums were renewed every two days.

(2) Cell proliferation by CCK-8 assays: According to the manufacture’s protocols, the proliferation of L929 cells onto the sample was determined by CCK-8 assays. After incubating for 1, 3 and 5 days, the cell-scaffold constructs were removed to a new petri dish to get rid of cells that did not adhere onto the scaffolds. Then, a fresh cell culture medium with 10 vol% Cell Counting Kit-8 (CCK-8, Dojindo, Kumamoto, Japan) solutions was added to incubate for another two hours. 200 μL of the reacted medium culture was measured at 450 nm by a microplate reader (SPECTRAmax 384, Molecular Devices, San Jose, CA, USA). As control groups, L929 cells were also seeded onto the petri dishes directly without scaffolds. Three samples were performed for each group.

(3) Cell adhesion and distribution by CLSM observation: A confocal laser scanning microscope (CLSM, Nikon A1R, Japan) was applied to determine the adhesion and distribution of fibroblasts in the PCL scaffolds. First, the cell-scaffold constructions were collected from the culture medium and then washed twice by using a phosphate buffer saline (PBS, HyClone, Logan, UT, USA). Then, the constructions were fixed by 4 vol% paraformaldehyde for 30 min, and washed three times by using PBS at room temperature. After that, the constructions were incubated in Phalloidine solution (Sigma-Aldrich, St. Louis, MO, USA) (5 μg/mL) for 30 min, and again washed three times with PBS. The samples were then incubated within the DAPI solution (Sigma-Aldrich, St. Louis, MO, USA) (2 μg/mL) for another 5 min. Finally, the samples were thoroughly washed with PBS before CLSM observation.

## 3. Results and Discussion

### 3.1. Thermal Analyses

First, DSC measurements have been carried out to investigate the thermal behaviors of PCL samples under non-isothermal conditions. Figure 2 shows the DSC thermograms of PCL samples, and Table 2 shows their thermal properties.

It is clear that samples with a low molecular weight (MW) (PCL-1 & PCL-2) show lower melting temperature, but higher crystallization temperature than their higher-MW counterparts (Figure 2), indicating that the increasing of MW can delay both the melting and crystallization of PCL samples. On the one hand, the lower MW samples contain more chain ends, which serve as lattice defects in the large crystal to lower their melting points [34]. On the other hand, the lower MW samples also possess better mobility, which can boost the fold of polymer chains into crystals to achieve an earlier and more rapid crystallization. Meanwhile, the crystallinity of each sample has been calculated as the quotient of the heat of fusion and the heat of fusion of 100% crystalline PCL [35]. As shown in Table 2, the higher-MW samples exhibit significantly lower crystallinity. The lower crystallinity of a high-MW sample might be ascribed to its relatively lower crystallization rate. With the chain length increasing, the effective diffusion on the crystal growth interface is slowed down, hence performing a longer crystallization half time (*t*_1/2_), and facilitating the generation of smaller sets of crystalline sections [36,37]. Basically, the low-MW samples achieve more rapid and earlier crystallization, which may promote the solidification of the pore structure in the following melt-state foaming process.

### 3.2. Rheological Behaviors of PCL Samples

To acquire the information on the rheological behaviors of PCL samples, the complex viscosity (*η**), storage modulus (*G*′) and loss modulus (*G*″) curves have been measured by frequency sweep tests at 80 °C. Figure 3 shows these characteristics as a function of frequency. 

Figure 3a shows that, in comparison with low-MW (PCL-1 & PCL-2) samples, the high-MW samples (PCL-3 & PCL-4) show remarkably higher complex viscosity (*η**), which is in agreement with previous studies [38]. Besides, the complex viscosity of low-MW samples hardly shows any shear-thinning phenomenon over the whole frequency, indicating similar characteristics to Newtonian fluids. It indicates that fewer entanglements between macromolecular chains exist in PCL-1 and PCL-2, which can be ascribed to their low MW. As shown in Figure 3b, high-MW samples also possess a higher loss modulus (*G*″) and storage modulus (*G*′) than low-MW samples, which is in agreement with the change of complex viscosity. Furthermore, Figure 3b shows that for low-MW samples (PCL-1 & PCL-2), the loss modulus (*G*″) is much higher than the storage modulus (*G*′) in the whole sweeping region, indicating that the value of *G*″/*G*′ of these two samples is much more than 1. Park et al. have investigated the relationship of pore stability and *G*″/*G*′, and demonstrate that samples tend to possess low pore stability and foamability when the value of *G*″/*G*′ is over 1 [39]. Therefore, in terms of the value of *G*″/*G*′, low-MW samples seem to possess poor foamability. Overall, compared with low-MW samples, high-MW samples gain higher complex viscosity, *G*′ and *G*″, indicating a higher melt strength. More importantly, the increased melt strength can provide enhanced resistance to the deformation of pores, including their growth and rupture.

### 3.3. Porous Morphology of PCL Scaffolds

Based on the above analyses, the selected four PCL samples show varied thermal and rheological properties, which may be a favorable aspect for the morphological optimization of PCL scaffolds by providing rich choices. Previous studies have shown that for PCL, foaming at melt state, mild pressure and rapid depressurization are beneficial to acquire a favorable interconnected and macroporous structure [22,23,24,25,27]. Lian, Z. et al. has reported that the −dTm/dP of PCL-CO_2_ system within 10 MPa is 2.02 °C/MPa, according to the Clapeyron equation [40]. Additionally, our previous work has demonstrated that once the foaming temperature is over 60 °C, the PCL exhibits poor foamability [27]. Therefore, in order to achieve desired morphology and decent foamability, the melt-state foaming has been carried out at the following foaming conditions: T = 50 °C, P = 10 MPa and venting time = 0.3 s. After foaming, the porous morphology of PCL scaffolds has been investigated by SEM and pore size, as well as porosity and interconnectivity measurements. The results are shown in Figure 4 and Table 3, respectively.

Notably, Figure 4 shows that four samples show clearly distinct porous morphology, indicating that the intrinsic properties of PCL samples have a significant effect on the foaming results. Quantitatively, PCL-1 possesses the largest pore size of 125 μm and sufficient interconnectivity of 96 % (Table 3). The lowest complex viscosity of PCL-1 provides the weakest resistance to the growth and rupture of pores, achieving favorable highly interconnected macropores, as a result. However, the rheological analyses also indicate that PCL-1 might have poor foamability for its high value of *G*″/*G*′. Delightfully, the early and rapid crystallization of PCL-1 enhances its foamability, since the early formed crystals can suppress the excessive collapse of porous structure [41]. Furthermore, Table 3 also shows that with the increase of MW, PCL-2 and PCL-3 exhibit the gradually decreased pore size and interconnectivity. Particularly among the four samples, PCL-3 gains the smallest pore size and interconnectivity of 43 μm and 82%, respectively. Previous crystallization and rheological tests demonstrate that compared with PCL-1, PCL-3 and PCL-2 exhibit gradually increased complex viscosity and delayed crystallization. The delayed crystallization can delay the solidification of PCL samples to favor the increase of pore size and interconnectivity. On the contrary, the remarkable increase of complex viscosity can conversely retard the expansion and rupture of porous structure, and as a result, tends to decrease pore size and interconnectivity. Here, the dominating effects of the increase of complex viscosity may contribute to the gradual reduction of both pore size and interconnectivity in PCL-2 and PCL-3. For PCL-1, PCL-2 and PCL-3, the increase of complex viscosity is about one order of magnitude between adjacent samples within 0.1–1000 rad/s (Figure 3a), while the increase of the crystallization temperature is only about 0.8 °C (Table 2), thus resulting in the dominating effect being controlled by the increase of complex viscosity. Interestingly, with the further increase of MW, PCL-4 gains a relatively bigger pore size, interconnectivity and porosity than PCL-3, which conflicts with previous observations in PCL-1, PCL-2 and PCL-3. The reason for the reversal results might be that the dominating effects on porous structure are switched from the increase of complex viscosity to the delay of crystallization. This is because between PCL-4 and PCL-3, the increase of complex viscosity is weakened, while the delay of crystallization is enhanced. As shown in Table 2 and Figure 3a, the increase of complex viscosity between PCL-4 and PCL-3 (3 folds to 1.4 folds) is about a quarter of that between PCL-3 and PCL-2 (12 folds to 4 folds) within 0.1–1000 rad/s, while the decrease of the crystallization temperature between PCL-4 and PCL-3 (1.4 °C) is nearly two folds of that between PCL-3 and PCL-2 (0.8 °C). Hence, the effect of delayed crystallization is significant. Overall, based on the foaming results and above analyses, we can safely conclude that the combined effects of rheological and crystallization properties modulate the porous morphology of PCL scaffolds. Specifically, PCL-1 possess favorable morphology (pore size = 70–180 μm, porosity = 90% and interconnectivity = 96%) as tissue engineering scaffolds, as the weakest melt benefit the generation of highly interconnected macropores, and the earliest crystallization maintain its foamability by suppressing the excessive collapse.

### 3.4. Compressive Properties of Porous PCL Scaffolds

A typical compressive testing has been performed by a universal testing machine to measure the strength and Young’s modulus of four foamed PCL scaffolds. Results are shown in Figure 5. It is clear that PCL-3 possess the highest strength among all samples, while PCL-4 obtains the lowest one. The varied compressive strength of PCL scaffolds is ascribed to their different porous morphology. PCL-3 possesses a relatively closed porous structure (lowest interconnectivity) and medium porosity, which is beneficial to obtain high strength. On the contrary, PCL-4 shows the highest porosity and high interconnectivity, to provide the weakest mechanical support. As for Young’s modulus, as shown in Figure 5c, PCL-1 shows the highest value, and the Young’s modulus of all samples decreases with the MW increasing. As mentioned in the DSC tests, the crystallinity of samples decreases, with the MW increasing. Therefore, the highest Young’s modulus of PCL-1 might be attributed to its high crystallinity. More importantly, previous studies have also reported that the improved stiffness is a beneficial aspect in terms of hard tissue engineering [42].

### 3.5. Biocompatibility of Porous PCL Scaffolds

To test the cellular responses to PCL-1 and PCL-3 scaffolds, CCK-8 assays and CLSM tests have been applied to measure the L929 cell proliferation and cell adhesion and distribution after seeding onto the scaffolds. As a control group, cells have also been directly cultured onto the petri dishes without scaffolds. The corresponding results are shown in Figure 6 and Figure 7, respectively.

Figure 6 shows the obvious proliferation of L929 cells in all groups for the first five days, which demonstrates that these cells have vigorous viability. Meanwhile, the cell number between PCL-1 and the control group appears no statistically different, except for on the 1st day (*p* < 0.05). On this day, the number of cells attached to the samples is approximately 80% of those attached to petri dishes. The macroporous structure of PCL-1 scaffold contributes to its relatively lower seeding efficiency, which is also found in a number of reports [43,44]. Notably, the similar proliferation trend between PCL-1 and the control group demonstrates that PCL-1 scaffolds provide a favorable microenvironment for cell proliferation. When compared with cells on the PCL-1 scaffolds, the proliferation of cells onto PCL-3 scaffolds is significantly slowed down in the 5th day, indicating that the insufficient pore size and interconnectivity of PCL-3 can hinder the further proliferation of seeded cells once the cell amount reaches a relatively large value. Furthermore, form Figure 7, the 2D CLSM images show that cells onto both the PCL-1 and PCL-2 scaffolds possess observable cell proliferation, which is consistent with the CCK-8 assay results. Vivid cytoplasmic extension and cellular communication are established after the 3rd and 5th days, indicating the perfect adhesion of cells. More importantly, the 3D CLSM images show that cells on the PCL-1 scaffolds can penetrate to a depth of over 200 μm, but the ingrowth of cells onto the PCL-3 scaffold is hindered on the 5th day. The distinct ingrowth capability of cells onto the PCL-1 and PCL-3 scaffolds is attributed to their different porous structure. The sufficient interconnectivity of PCL-1 scaffolds provides essential channels for the ingrowth of cells, while the pores in PCL-3 scaffolds are too small and closed for the penetration of cells. Overall, both the CCK-8 and CLSM assays confirm that the PCL-1 scaffold with proper porous morphology performs excellent biocompatibility for potential regenerative applications. 

## 4. Conclusions

In this work, we have prepared PCL scaffolds via efficient scCO_2_ melt-state foaming, and studied the relation between the porous morphology of samples and their thermal and rheological properties. On one hand, the early and rapid crystallization promotes the solidification of the porous structure by suppressing the growth and rupture of pores, and helps them to maintain foamability. While on the other, the decrease of complex viscosity is beneficial to the deformation of pores namely the growth and rupture of pores. As a result, the porous morphology of PCL scaffolds is modulated by the net effects of crystallization and rheological properties. Remarkably, PCL-1 scaffolds with the weakest melt and the earliest crystallization are ascertained to possess the proper intrinsic properties for the fabrication of favorable highly interconnected and macroporous morphology (pore size = 70–180 μm, porosity = 90% and interconnectivity = 96%). Meanwhile, PCL-1 also provides the highest stiffness, owing to its high crystallinity. Additionally, the developed structure in PCL-1 scaffolds provides a favorable microenvironment for cellular proliferation, adhesion and ingrowth, which is promising for regenerative applications. To conclude, the elaborate selection of intrinsic properties of PCL samples like rheological and crystallization behaviors is the key to obtain desired foam morphology.

## Figures and Tables

**Figure 1 polymers-12-00780-f001:**
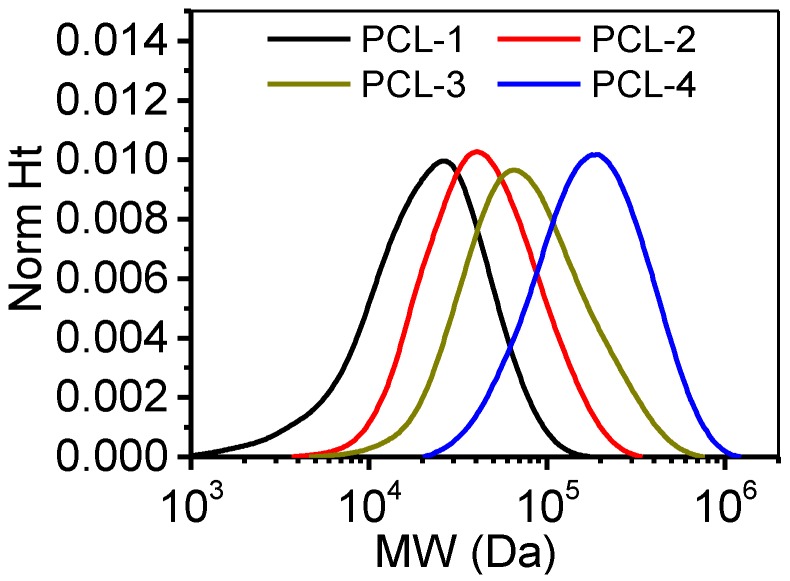
Molecular weight (MW) distribution of PCL samples according to GPC measurements.

**Figure 2 polymers-12-00780-f002:**
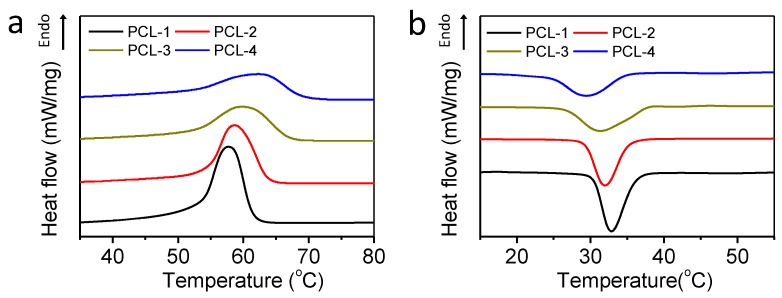
DSC thermograms of PCL samples with different molecular weight; (**a**) heating process; (**b**) cooling process.

**Figure 3 polymers-12-00780-f003:**
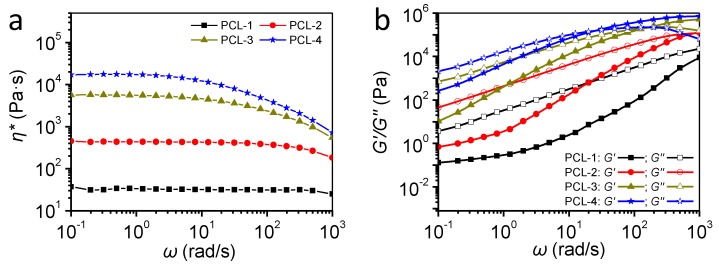
Rheological behaviors of PCL with different molecular weight; (**a**) complex viscosity as a function of frequency; (**b**) storage and loss modulus as a function of frequency.

**Figure 4 polymers-12-00780-f004:**
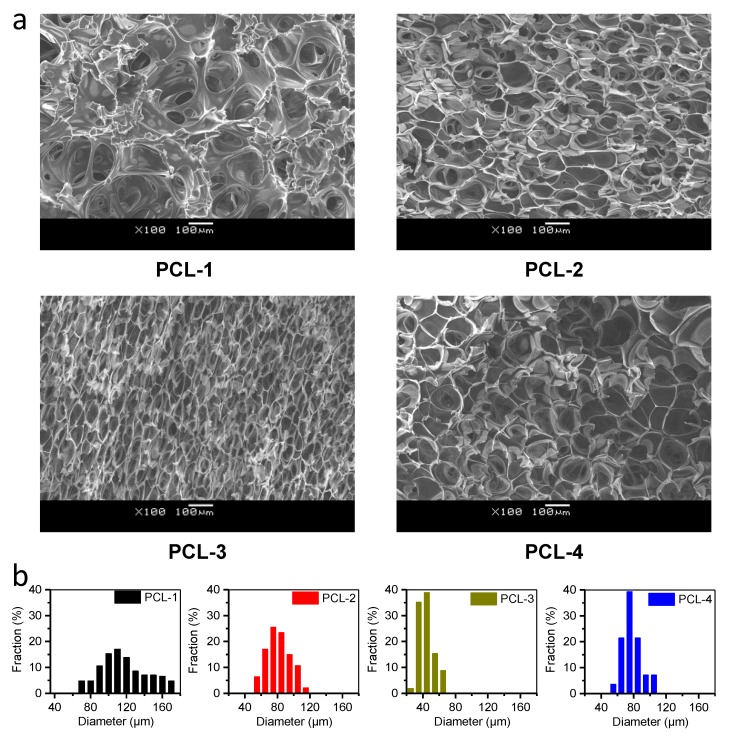
SEM images (**a**) and pore size distribution (**b**) of PCL scaffolds prepared at 50 °C, 10 MPa and a venting time of 0.3 s.

**Figure 5 polymers-12-00780-f005:**
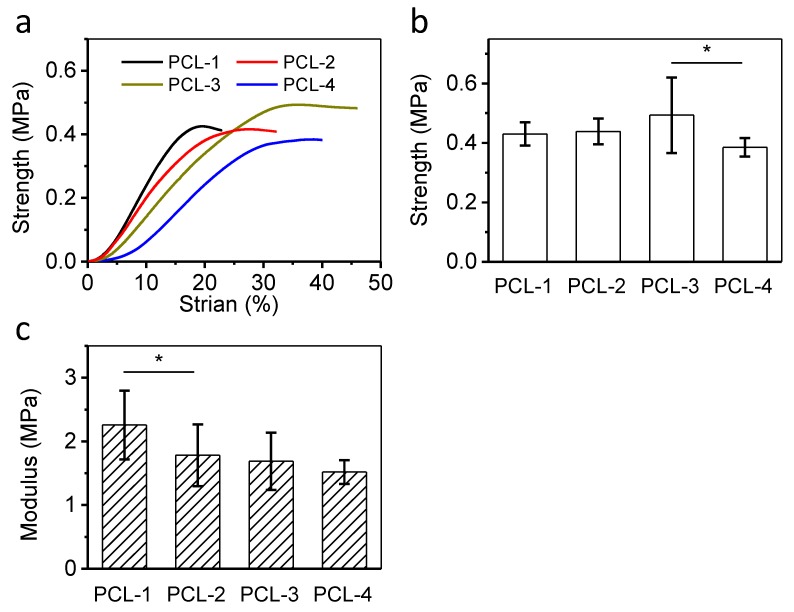
Compressive properties of the PCL scaffolds. (**a**) The stress-strain curves; (**b**) their yield strengths; and (**c**) Young’s modulus at 10% strain (* *p* < 0.05).

**Figure 6 polymers-12-00780-f006:**
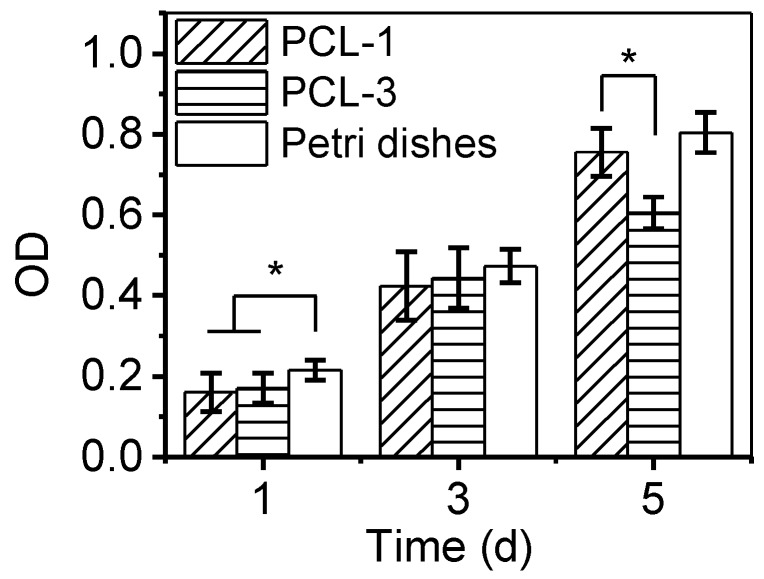
CCK-8 assay results of L929 cells seeded onto the PCL-1 and PCL-3 scaffolds on the 1st, 3rd and 5th day. The petri dishes are served as the control group for comparison. All values are the means with ± SD. *n* = 3 (**p* < 0.05).

**Figure 7 polymers-12-00780-f007:**
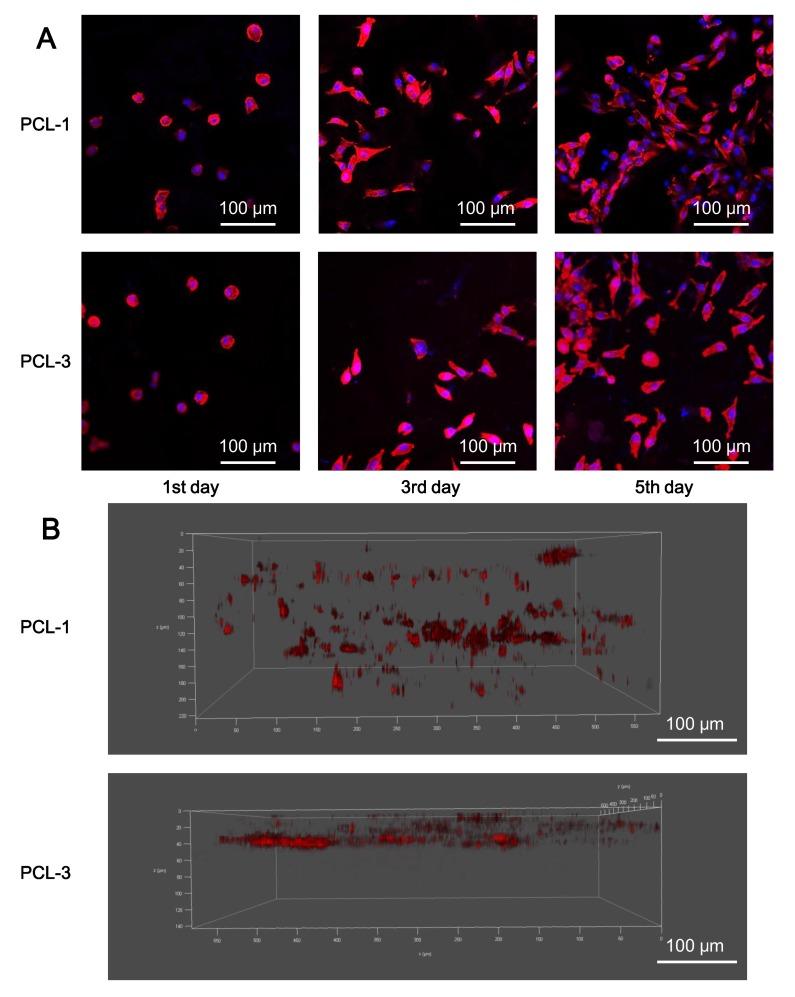
CLSM images showing the adhesion and distribution of cells seeded onto the PCL-1 and PCL-3 scaffolds. (**A**) 2D CLSM images of L929 cells seeded onto the scaffolds on the 1st, 3rd and 5th day; and (**B**) 3D CLSM images of the distribution L929 cells onto the scaffolds in the 5th day.

**Table 1 polymers-12-00780-t001:** GPC results of different PCL samples.

Samples	*M_w_*	*M_n_*	*PDI*
PCL-1	27 kDa	14 kDa	1.876
PCL-2	54 kDa	32 kDa	1.688
PCL-3	100 kDa	56 kDa	1.781
PCL-4	219 kDa	137 kDa	1.604

**Table 2 polymers-12-00780-t002:** Thermal properties of the PCL with different molecular weight.

	*T_m_* (°C)	*T_c_* (°C)	*X_c_* (%)	*t_1/2_* (min)
PCL-1	57.8	33.1	53	0.39
PCL-2	58.9	32.2	44	0.49
PCL-3	60.2	31.4	40	0.70
PCL-4	63.0	30	38	0.78

**Table 3 polymers-12-00780-t003:** Quantitative analyses of porous morphology of PCL scaffolds (prepared at 50 °C, 10 MPa and a venting time of 0.3 s) and their crystallization and rheological properties.

Samples	Complex Viscosity	Crystallization		Porous Morphology		
		Rate	Temperature	Average Pore Size (µm)	Interconnectivity (%)	Porosity (%)
PCL-1	Lowest	Rapid	High	125	96	91
PCL-2	low	Medium	medium	73	91	91
PCL-3	Medium	Slow	Low	43	82	92
PCL-4	High	Slowest	Lowest	82	84	94

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
