# Peer review of "Fabrication of PCL Scaffolds by Supercritical CO2 Foaming Based on the Combined Effects of Rheological and Crystallization Properties"

_polymers, 2020, doi:10.3390/polym12040780_

Round 1

Reviewer 1 Report

The paper aims at correlating the morphology of PCL scaffolds with their rheological and crystallization behaviors to determine the proper intrinsic properties that affect the preparation of a highly interconnected and macroporous scaffolds

Comments:

  • The article is generally written well but needs a second look at some of the grammatical errors and correcting few typos and, for example:
    • The sentence on line 18 starting with “Herein, in  this  study,  ….) remove herein as it has the same meaning of in this study
    • The paragraph on line 97 starting with “In current study, we tried to correlate…better to write “ In this study…
    • Line 114… approximate should be …increase to approximately two folds. The same comment applies to line 129

  • The article has a high degree of similarity in writing and wording upon doing iThenticate analysis with the reference “Controllable fabrication of porous PLGA/PCL bilayer membrane for GTR using supercritical carbon dioxide foaming” of same authors. Please try to write in a new style and wording to avoid plagiarism.

  • The introduction needs to refer to more recent references and updated other methods to compare with for preparations of the ideal scaffold. For example, the authors failed to mention and compare their approach to the recent reactive electrospinning approach which utilize less harsh conditions and uses compatible or safe solvents, such as the study reported by Ismail et al. in Polymers 10(4), 455 (2018) to prepare electrospun nanofibrous scaffolds with superior pore size, porosity, and interconnectivity

  • Under the methodology of GPC, more details should be provided regarding the refractive index methodology and what differential refractive index increment (dn/dc) used?

  • Why glass transition temperatures for the PLL samples were not reported knowing that this change could affect chain mobility and, subsequently, the polymer crystallinity and interaction with living cells?
  • Mechanical properties of the prepared scaffolds must be conducted as this is one of the determining factors that decide the use and durability of any scaffolds and in which application. It is essential to supplement the lack of enough characterization in this paper.

Reviewer 2 Report

The authors study the role of the rheology and the crystallinity on the production of PCL foams with scCO2 for biomedical applications. They also took into account the effect of the PCL molecular weight on scaffolds pore size and interconnectivity and the authors were able to explain their results by considering their previous PCL characterizations. The article is well organized, and also provides interesting results. However, it must be completely rewritten in terms of english and some results have to be explained more precisely. Also, there is a lack of explanations in the methodology section.

English is very bad and some words and expressions must be corrected. µm instead of um, “rapidest”, melting point instead of melt point, “famability”,… Melt strength is a term that is used concerning extensional viscosity and the stress that can be applied to a melt without breaking. The article refers to melt strength regarding melting temperature. The term is completely wrong and must be corrected. Authors performed a frequency sweep. But, how are they sure that they are performing that experiment in the linear viscoelastic region? They should give the deformation at which they are performing the experiments. Authors perform the rheological tests at 80 ºC. However, according to the DSC experiments, they are working above the melting point. I am not sure that they would obtain the same results under PCL melting point. Temperature sweep experiments are needed to rheologically study the different samples. Authors performed the experiments at 100 bar and 50 ºC. Although the conditions are not really hard, it is expected a decrease in the melting point due to the effect of the pressure. Melting point depression phenomenon must be considered and must be studied previously because important changes can be produced. Please, provide the GPC chromatograms to determine the molecular weight distributions. It is true that the authors provide the PDI, but the figures are also needed. How did the authors perform the image analysis? They should explain if they are counting the pores, the number of the pictures they used to do that, number of pores they counted, or if they are using a threshold modeling. How many pores are counting (pore histograms are in fractions). More information is needed concerning this issue. How did the authors determine the cell adhesion? It is not explained in the manuscript. Cell proliferation measurement is explained, but not cell adhesion. Both things are different, and cells can proliferate properly, without a correct cell adhesion percentage. Authors selected PCL1, based on the results of pore size and interconnectivity. I agree that they are perhaps the most important properties for cells. However, I believe that this experiments must be performed with another sample to study if, in this case, these phenomena are the key for this foam.

Round 2

Reviewer 1 Report

-The transcript should be again checked well for typos and grammatical errors. Here are some of the typo’s examples:

  • Line 48, It should be corrected to photocrosslinked electrospinning method .. (not electrospunning method)
  • Line 50, the sentence should be written as ….about ten microns, which may have negative… (not should have)
  • Line 59, should be…still a big challenge to fabricate stable PCL scaffolds …..(not stably fabricate)
  • Line 336, remove then…A typical compression testing…

- The changes made to previous comments were addressed satisfactorily

Reviewer 2 Report

Authors answered reviewer´s comments in a satisfactory way. The article can be published.

Author Response

Thank you for the kind comments